# Orbitofrontal cortex contributes to the comparison of values underlying economic choices

Sébastien Ballesta [1,4,5], Weikang Shi [1,6] & Camillo Padoa-Schioppa [1,2,3] ✉

Economic choices between goods entail the computation and comparison of subjective values. Previous studies examined neuronal activity in the orbitofrontal cortex (OFC) of monkeys choosing between different types of juices. Three groups of neurons were identified: offer value cells encoding the value of individual offers, chosen juice cells encoding the identity of the chosen juice, and chosen value cells encoding the value of the chosen offer. The encoded variables capture both the input (offer value) and the output (chosen juice, chosen value) of the decision process, suggesting that values are compared within OFC. Recent work demonstrates that choices are causally linked to the activity of offer value cells. Conversely, the hypothesis that OFC contributes to value comparison has not been confirmed. Here we show that weak electrical stimulation of OFC specifically disrupts value comparison without altering offer values. This result implies that neuronal populations in OFC participate in value comparison.

Recent work demonstrated that offer values encoded in OFC are causal to choices[1]. In contrast, where in the brain and how exactly values are compared to make a decision remains an open question. Several hypotheses have been put forth. When monkeys choose between goods, different groups of cells in OFC encode the choice input (offer values) and the choice outcome (chosen good, chosen value)[2,3], suggesting that the cell groups identified in this area constitute the building blocks of a decision circuit. Experimental findings[4,5] and computational models[6–8] support this hypothesis, but the evidence remains correlative. Other studies suggested that economic decisions take place in motor systems[9,10], through distributed processes[11,12], through shifts of visual attention[13], by integrating hippocampal signals[14], or without the explicit comparison of values[15]. However, none of these proposals has been validated by causal evidence. The present study specifically examined whether OFC contributes to value comparison.

We introduce a paradigm to assess whether a neural population contributes to a decision. At the neural level, a binary decision is ultimately a comparison between two neural signals. For example, in the random dot task used to study the visual perception of motion[16], the decision is a comparison between two neural signals representing motion in the two valid directions. Conversely, in economic choices, the decision is a comparison between two neural signals representing the offer values. In general, the signals to be compared are the input of the decision process. Now consider an experiment in which altering the activity of a neuronal population induces higher choice variability (shallower psychometric curves). In principle, an increase in choice variability may be due (a) to noisier (more ambiguous) input signals, (b) to a noisier decision process, or (c) to the disruption of the subsequent motor response. If one can exclude (a) that the manipulation makes input signals more noisy and (c) that it disrupts motor planning, one may conclude (b) that the neural population participates in the decision process. We now consider three previous results in this light.

First, classic studies showed that low-current stimulation (≥10 μA) of the middle temporal area biased perceptual decisions[16,17]. Furthermore, high-current stimulation (≥40 μA) increased choice variability[18].

[1]Department of Neuroscience, Washington University in St. Louis, St. Louis, MO 63110, USA. [2]Department of Economics, Washington University in St. Louis, St. Louis, MO 63110, USA. [3]Department of Biomedical Engineering, Washington University in St. Louis, St. Louis, MO 63110, USA. [4]Present address: Laboratoire de Neurosciences Cognitives et Adaptatives (UMR 7364), Strasbourg, France. [5]Present address: Centre de Primatologie de l'Université de Strasbourg, Niederhausbergen, France. [6]Present address: Department of Neuroscience, Yale University, New Haven, CT 06510, USA. ✉e-mail: camillo@wustl.edu

The latter observation was interpreted as follows. At low current, stimulation affected only neurons within one minicolumn, resulting in a bias. At high current, stimulation also affected neurons in other minicolumns, with opposite preferred direction[19]. This increased ambiguity and lead to higher choice variability. In the language adopted here, the increase in choice variability was due to noisier input signals.

Second, experiments in mice found that optogenetic inactivation of OFC increased choice variability, and that this effect was due to animals reverting to stereotyped behavior[20]. In other words, when OFC was inactive, mice "chose" by adopting strategies such as selecting the same juice chosen in the previous trial or consistently licking on one side. These results indicated that OFC is necessary for economic choices. However, the results did not disambiguate whether OFC's role is in value assignment, value comparison, or both.

Third, in our recent experiments, monkeys chose between two juices offered sequentially[1]. Electrical stimulation was delivered during presentation of the first offer (offer1) or the second offer (offer2) at 25, 50, or 125 μA (in separate sessions). We measured three behavioral effects: range-dependent bias, changes in order bias, and increase in choice variability. Both range-dependent bias and changes in order bias were interpreted as due to the stimulation altering offer values by increasing the activity of offer value cells (input signals). These biases were observed in different ways at all current levels ≥25 μA (see "Discussion"). Most relevant here, stimulation at 125 μA during offer2 but not during offer1 induced an increase in choice variability. Since the decision was made only upon presentation of offer2, it was tempting to interpret the last results as evidence that the stimulation interfered with value comparison and that this process took place within OFC. However, high current stimulation could affect fibers in the white matter adjacent to the stimulation site[21] and thus disrupt transmission to or between other brain areas[22]. Since the increase in choice variability was measured only at 125 μA, one cannot exclude that the decision process took place in some other brain region and/or that stimulation disrupted the early stages of motor planning [23].

Ideally, to provide evidence that OFC contributes causally to value comparison, one would need some condition in which weak electrical stimulation increases choice variability but does not induce any range-dependent bias or change in order bias. In other words, the stimulation should disrupt value comparison without altering offer values per se. In such a condition, the stimulation weakness would ensure that its effects are confined to OFC[21]. Furthermore, the absence of spatial or motor signals in this area[2,24,25] would ensure that the stimulation does not disrupt motor planning per se. Thus one could conclude that OFC participates in value comparison. In the experiments described below, we identified such a condition.

## Results

### Weak electrical stimulation of OFC disrupts value comparison

In each experimental session, a monkey chose between two juices labeled A and B, with A preferred, offered in variable amounts. The two offers were presented sequentially in the center of the monitor (Fig. 1A). Trials in which juice A was offered first and trials in which juice B was offered first were referred to as "AB trials" and "BA trials", respectively. The terms "offer1" and "offer2" referred to the first and second offer, independent of the juice type and amount. For each pair of juice quantities, the presentation order (AB, BA) and the spatial location of the saccade targets varied pseudo-randomly and were

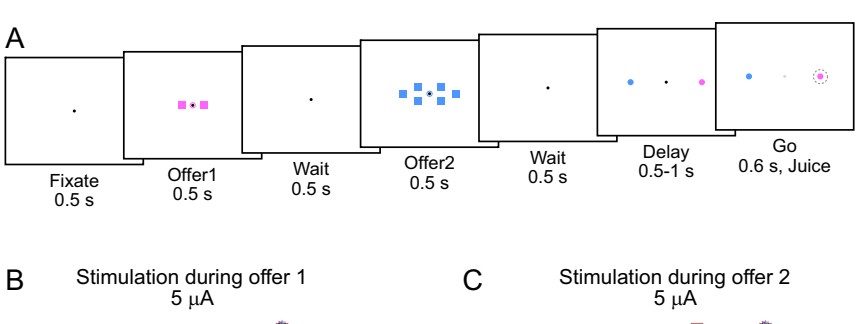

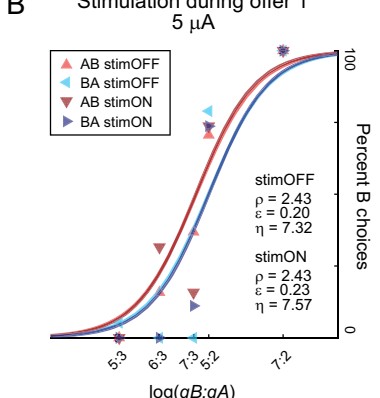

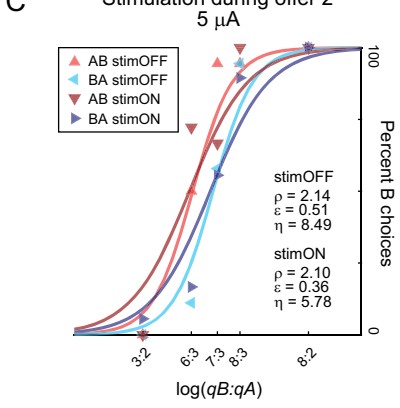

**Fig. 1 | Experimental design and example sessions. A** Trial structure. Each trial began with the animal fixating a dot (0.35° of visual angle). After a brief delay, two offers appeared in sequence, interleaved by a wait time. Each offer was represented by a set of colored squares (side = 1° of visual angle); the color indicated the juice type and the number of squares indicated the juice amount. Along with the offer, a small colored circle (0.75° of visual angle) appeared around the fixation dot. The circle indicated to the animal the juice identity in the case of null offer (0 drops; forced choices). The animal maintained center fixation throughout the trial until the go signal, indicated by the extinction of the fixation dot. The animal indicated its choice with a saccade and maintained peripheral fixation for an additional delay before juice delivery. Center fixation was imposed within 3°. Weak electrical stimulation (5–15 μA) was delivered during offer1 presentation or during offer2 presentation, in separate sessions. **B, C** Example sessions. Each panel represents one session. Red and blue refer to AB and BA trials, respectively; light and dark colors refer to stimOFF and stimON, respectively. Sigmoids were obtained from probit regressions. For stimOFF trials and stimON trials, the order bias captured the distance between the red and blue sigmoids. Weak electrical stimulation was delivered during offer1 (5 μA; panel **A**) or during offer2 (5 μA; panel **B**). Stimulation during offer1 did not substantially alter any of the choice parameters ($\rho$, $\varepsilon$, $\eta$). Similarly, stimulation during offer2 did not substantially alter the relative value ($\rho$) or the order bias ($\varepsilon$). However, stimulation during offer2 significantly decreased the sigmoid steepness ($\eta$). Source data are provided as a Source Data file.

counterbalanced across trials. Sessions typically included ~400 trials and offered quantities varied from trial to trial pseudo-randomly. An "offer type" was defined by two juice quantities in given order (e.g., [1A, 3B] or [3B, 1A]). To discourage monkeys from making a decision prior to offer2, we designed offer types such that for most values of offer1 the animal split choices between the two offers [26].

Weak electric current was delivered during offer1 or during offer2, in separate sessions. In each session, stimulation was delivered in half of non-forced choice trials, pseudo-randomly selected (see "Methods"). Compared to previous work using electrical stimulation of OFC[1], the main difference is that in the present study we used very low currents, in the range 5–15 µA.

For each session, we examined separately trials with and without the stimulation (stimON, stimOFF). For each group of trials, we analyzed choices with a probit regression:

$$\text{choice } B = \Phi(X)$$
$$X = a_0 + a_1 \log(q_B/q_A) + a_2(\delta_{\text{order},AB} - \delta_{\text{order},BA}) \tag{1}$$

where *choice B* = 1 if the animal chose juice B and 0 otherwise, $\Phi$ was the cumulative function of the standard normal distribution, $q_A$ and $q_B$ were the quantities of juices A and B offered, $\delta_{\text{order},AB} = 1$ in AB trials and 0 in BA trials, and $\delta_{\text{order},BA} = 1 - \delta_{\text{order},AB}$. From the fitted parameters, we derived measures for the relative value $\rho = \exp(-a_0/a_1)$, the sigmoid steepness $\eta = a_1$, and the order bias $\varepsilon = a_2$. Intuitively, $\rho$ was the quantity ratio $q_B/q_A$ that made the animal indifferent between the two juices, $\eta$ was inversely related to choice variability, and $\varepsilon$ quantified the order bias. Specifically, $\varepsilon < 0$ ($\varepsilon > 0$) indicated a bias in favor of offer1 (offer2).

Figure 1B, C illustrates the main results of this study. In one example session (Fig. 1B), weak stimulation (5 µA) was delivered during offer1. The stimulation did not substantially alter the relative value ($\rho$), the order bias ($\varepsilon$), or the sigmoid steepness ($\eta$). In another example session (Fig. 1C), weak stimulation (5 µA) was delivered during offer2. Again, the stimulation did not substantially alter the relative value ($\rho$) or the order bias ($\varepsilon$). However, electrical stimulation substantially decreased the sigmoid steepness ($\eta$). In other words, weak stimulation during offer2 selectively increased choice variability.

Similar patterns were observed at the population level. Our data set included $N = 49$ sessions in which weak electrical stimulation was delivered during offer1, and $N = 42$ sessions in which weak stimulation was delivered during offer2. In general, neither the relative value ($\rho$) nor the order bias ($\varepsilon$) were substantially altered by electrical stimulation in either time window (Fig. 2A, B). In contrast, the sigmoid steepness ($\eta$) was significantly reduced by stimulation delivered during offer2 but not during offer1 (Fig. 2C). That is, weak electrical stimulation during offer2 but not during offer1 significantly increased choice variability. This effect was observed in each of two animals (Supplementary Fig. 1).

**Weak stimulation of OFC does not induce reversion to stereotyped behavior**

We conducted several control analyses. First, we considered the possibility that the increase in choice variability measured at very low current might be driven by reduced motivation or by a generic disengagement from the task. Arguing against this view, electrical stimulation in either time window did not systematically increase the error rate (Fig. 3A), nor did it alter response times (Fig. 3B).

Second, previous work found that optogenetic inactivation of OFC in mice induced an increase in choice variability, and that this effect was due to animals reverting to stereotyped behavior[20]. We thus examined whether weak electrical stimulation of OFC in monkeys affected choices in similar ways. Other things equal, animals might have a choice bias favoring one side (side bias). Similarly, they might tend to choose on any given trial the same option chosen in the

previous trial (choice hysteresis)[4,27,28]. These biases−or stereotyped behaviors−would contribute to choice variability defined in Eq.1. Thus in a series of analyses, we examined whether electrical stimulation altered these biases. Of note, in our choice task, the tendency to repeat choices (choice hysteresis) could refer to the juice type, the target location, or the presentation order. Using probit regressions, we considered each potential bias in turn.

For the side bias, we examined each group of trials (stimOFF, stimON) with the following model:

$$\text{choice } B = \Phi(X)$$
$$X = a_0 + a_1 \log(q_B/q_A) + a_2(\delta_{A,\text{right}} - \delta_{B,\text{right}}) \tag{2}$$

where $\delta_{J,\text{right}} = 1$ if the target associated with juice J was presented on the right and 0 otherwise, and J = A, B. The side bias was defined as $\xi = a_2$. A measure $\xi > 0$ indicated that other things equal, the animal tended to choose the left target. Population analyses showed that the side bias was not substantially altered by weak stimulation of OFC in either time window (Fig. 4A). (If anything, stimulation during offer1 *reduced* the side bias.)

For choice hysteresis (juice type), we analyzed each group of trials (stimOFF, stimON) with the following model:

$$\text{choice } B = \Phi(X)$$
$$X = a_0 + a_1 \log(q_B/q_A) + a_2(\delta_{n-1,B} - \delta_{n-1,A}) \tag{3}$$

where $\delta_{n-1,J} = 1$ if in the previous trial the animal chose juice J and 0 otherwise, and J = A, B. Choice hysteresis was quantified as $\theta_{\text{juice}} = a_2$. A measure of $\theta_{\text{juice}} > 0$ indicated that, other things equal, the animal tended to choose the same juice chosen in the previous trial. Confirming previous reports[4], a population analysis revealed a consistent choice hysteresis related to the juice type (mean($\theta_{\text{juice}}$) > 0). Critically, weak stimulation of OFC in either time window did not systematically alter this phenomenon (Fig. 4B).

For choice hysteresis (side), we used the following model:

$$\text{choice } B = \Phi(X)$$
$$X = a_0 + a_1 \log(q_B/q_A) + a_2(\delta_{n-1,\text{sideB}} - \delta_{n-1,\text{sideA}}) \tag{4}$$

where $\delta_{n-1,\text{side J}} = 1$ if the target associated with juice J was in the same spatial position as that chosen in the previous trial and 0 otherwise, and J = A, B. Choice hysteresis was quantified as $\theta_{\text{side}} = a_2$. A measure of $\theta_{\text{side}} > 0$ indicated that, other things equal, the animal tended to choose the same saccade target as that chosen in the previous trial. Population analyses showed that weak stimulation of OFC in either time window did not systematically alter choice hysteresis related to the chosen side (Fig. 4C).

Finally, for choice hysteresis (order), we used the following model:

$$\text{choice } B = \Phi(X)$$
$$X = a_0 + a_1 \log(q_B/q_A) + a_2(\delta_{n-1,\text{orderB}} - \delta_{n-1,\text{orderA}}) \tag{5}$$

where $\delta_{n-1,\text{order J}} = 1$ if the order of presentation of juice J in the present trial was same of the juice chosen in the previous trial and 0 otherwise, and J = A, B. Choice hysteresis was quantified as $\theta_{\text{order}} = a_2$. A measure of $\theta_{\text{order}} > 0$ indicated that other things equal, the animal tended to repeat (as opposed to alternate) choices according to the presentation order. Population analyses showed that weak stimulation of OFC in either time window did not systematically alter choice hysteresis related to the presentation order (Fig. 4D).

In conclusion, the drop in choice accuracy induced by weak electrical stimulation delivered during offer2 was not due to the animals reverting to stereotyped behavior.

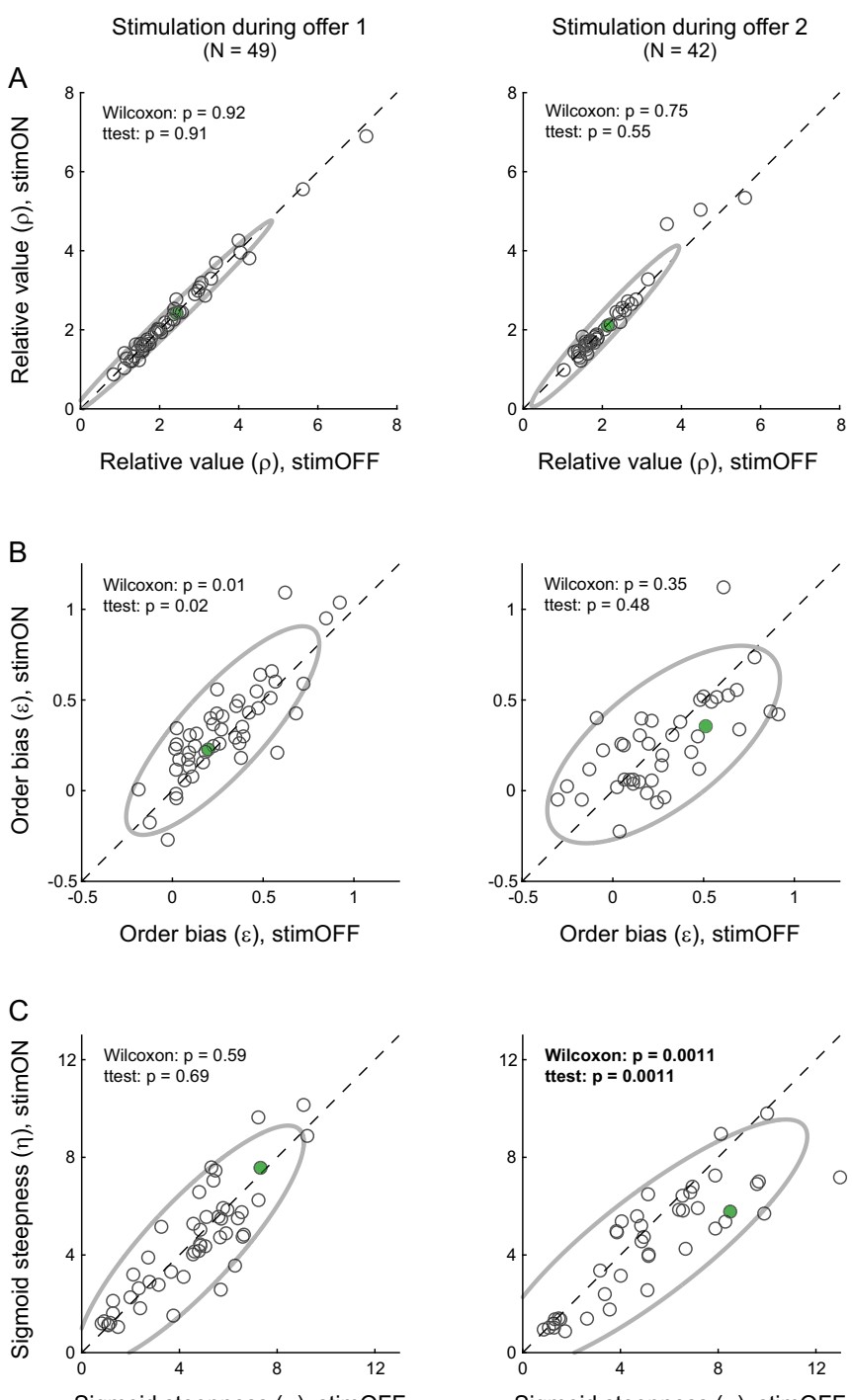

**Fig. 2 | Weak stimulation of OFC selectively disrupts value comparison.**
**A** Relative value ($\rho$). The two panels illustrate the effects of stimulation during offer1 and offer2, respectively. In each panel, *x*-axis and *y*-axis represent the relative value measured for stimOFF and stimON trials, respectively. Each data point represents one session and the ellipse represents the 90% confidence interval. Weak OFC stimulation during wither time window did not systematically alter the relative value. **B** Order bias ($\varepsilon$). Weak OFC stimulation during offer2 presentation did not affect the order bias. Stimulation during offer1 marginally increased this bias ($p = 0.02$, t test). **C** Sigmoid steepness ($\eta$). Weak OFC stimulation during offer2 but not during offer1 induced a significant increase in choice variability. Each panel reports the *p* values obtained from a two-tailed Wilcoxon test and a two-tailed t test. The two sessions illustrated in Fig. 1B, C are highlighted in green. Source data are provided as a Source Data file.

## Discussion

Electrical stimulation may increase choice variability by interfering with value computation and/or with value comparison. In our choice task, value computation took place during presentation of both offer1 and offer2, while value comparison took place only during presentation of offer2. Two lines of evidence suggest that weak electrical stimulation selectively affected value comparison. First, choice variability increased only when the stimulation was delivered during offer2. Second, the stimulation did not induce any order or range-dependent bias (i.e., it did not affect valuation per se). In principle, electrical stimulation could also increase choice variability by reducing the animal's motivation or engagement in the task. However, the absence of

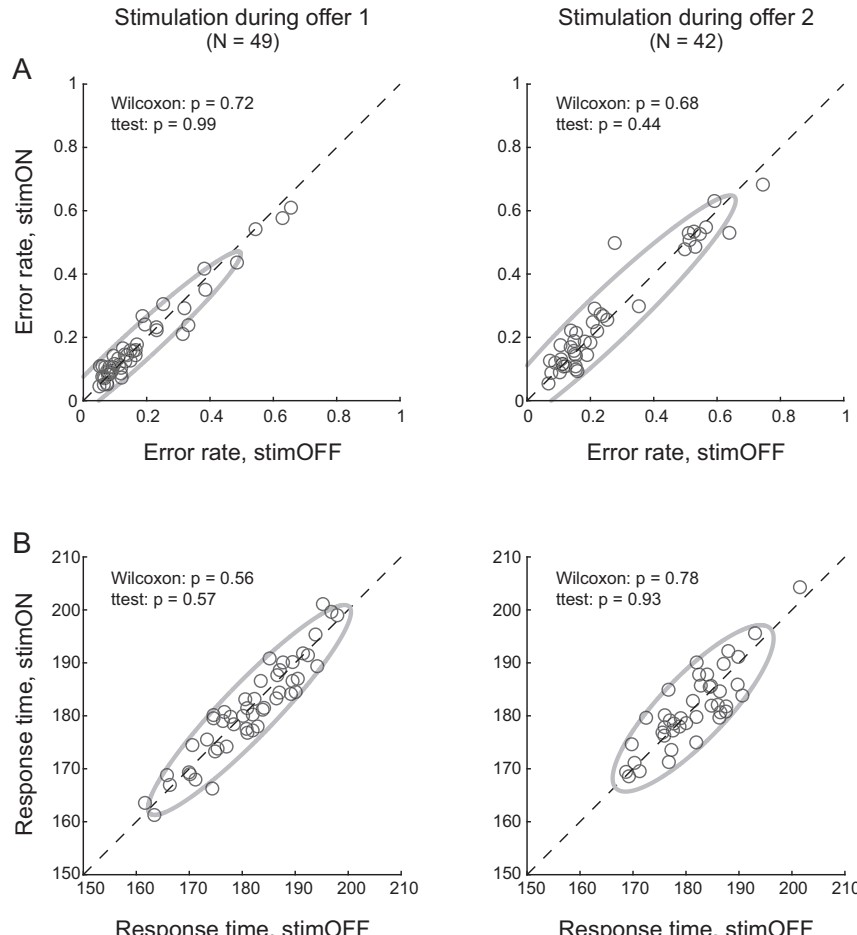

**Fig. 3 | Weak stimulation of OFC does not affect error rates and response times.**
**A** Error rate. Left and right panels illustrate the results obtained for stimulation at 5–15 μA during offer1 and offer2, respectively. In each panel, *x*-axis and *y*-axis represent the error rate measured for stimOFF and stimON trials, respectively. Each data point represents one session, the gray ellipse represents the 90% confidence interval, and the results of statistical tests are reported. The error rate (i.e., the fraction of trials that did not end in juice delivery) was not significantly altered by stimulation in either time window. **B** Response time. In each panel, *x*-axis and *y*-axis represent the mean response time measured for stimOFF and stimON trials, respectively. Other conventions are as in panel **A**. Response times were not significantly altered by stimulation in either time window. All *p* values were obtained from two-tailed Wilcoxon tests and two-tailed t tests. Source data are provided as a Source Data file.

any effect on error rates or reaction times argues against this hypothesis. Finally, stimulation could increase choice variability by disrupting the early stages of action planning. However, the primate OFC lacks spatial or motor representations[2,25], and stimulation at very low current is unlikely to affect neurons in other brain regions. In conclusion, our results indicate that neuronal activity in OFC contributes to value comparison.

In the present study, the increase in choice variability induced by weak stimulation of OFC was not due to the animals reverting to stereotyped behavior. In contrast, we previously observed that optogenetic inactivation of OFC in mice significantly increased stereotyped behavior (and thus increased choice variability)[20]. This apparent discrepancy may be interpreted considering that optogenetic inactivation, which was procured through excitation of inhibitory interneurons, had a dramatic effect on the activity of OFC neurons, as confirmed by simultaneous neuronal recordings. In essence, optogenetic inactivation shut down and thus "took offline" the target area. Absent the neural substrate normally devoted to the computation and comparison of offer values, animals' motor responses were presumably guided by alternative computations—i.e., stereotyped behaviors such as repeating on any given trial the same choice made in the previous trial. In this view, the mental processes guiding choices under optogenetic inactivation were qualitatively

different from those taking place under normal conditions. In contrast, the weak electrical stimulation used here presumably had a mild effect on neuronal activity in OFC, inducing noise or degrading the quality of value comparison without completely preventing neuronal computations from taking place. Thus, the mental processes guiding animals' choices were noisier, but not fundamentally different from those taking place under normal conditions.

It is interesting to contrast the results of weak electrical stimulation, described here, with those previously obtained with higher currents[1]. As noted above, we quantified three behavioral effects: range-dependent bias, changes in order bias, and increase in choice variability. To compare the results obtained with different protocols, we normalized the effect sizes (see "Methods"). Inspection of Fig. 5 reveals that three effects varied with the current level in different ways.

First, the range-dependent bias had a bell-shaped trend. All our measures are consistent with the understanding that electrical stimulation up to ~50 μA induced physiological increases in neuronal firing rates[21,29–31]. The tuning curves of offer value cells are linear and range adapting[32,33]. Thus increasing firing rates is equivalent to increasing each offer value by a quantity proportional to the value range. Hence, stimulation induced a choice bias favoring the juice offered with the larger range. This effect increased with the current level until 50 μA.

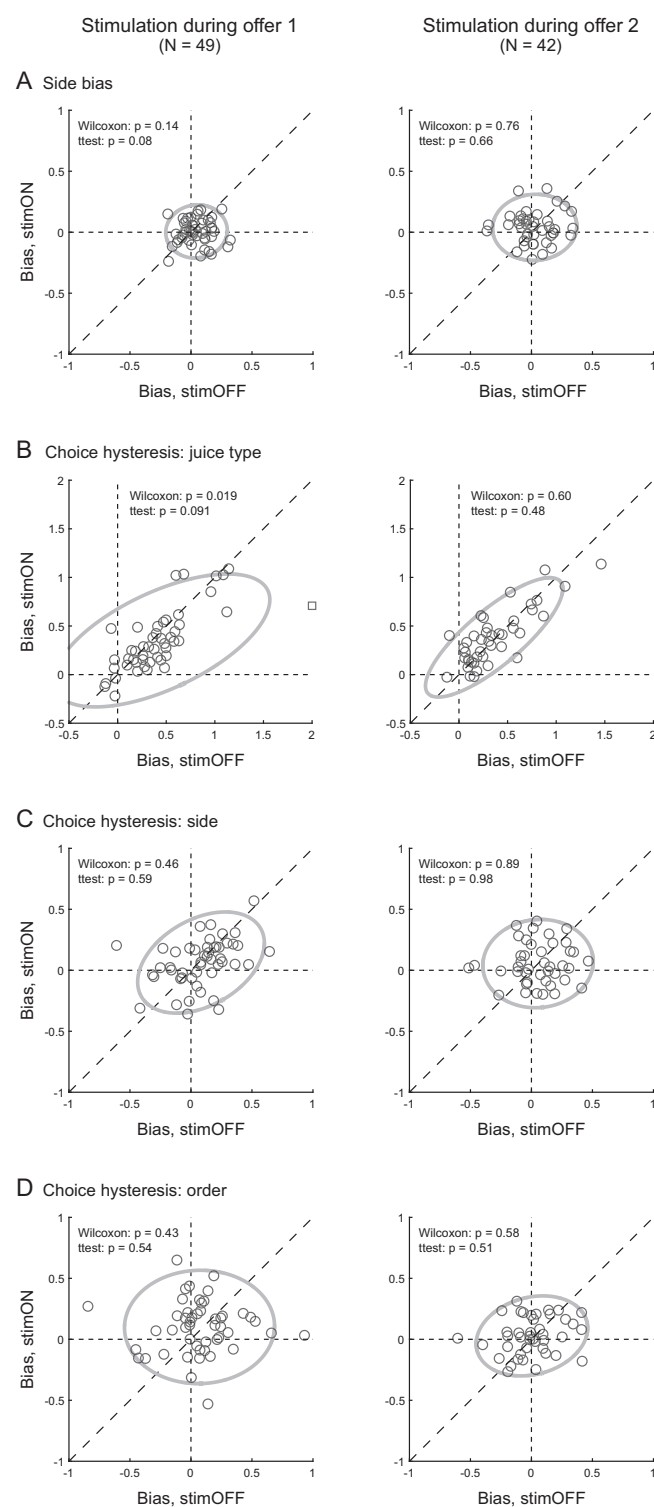

**Fig. 4 | Weak stimulation of OFC does not increase stereotyped behaviors.**
**A** Side bias. Left and right panels illustrate the results obtained for stimulation during offer1 and offer2, respectively. In each panel, *x*-axis and *y*-axis represent the side bias measured for stimOFF and stimON trials, respectively. Each data point represents one session and the gray ellipse represents the 90% confidence interval. In essence, the side bias was very modest, and electrical stimulation in either time window did not systematically increase it. **B** Choice hysteresis, juice type. Same conventions as in panel (**A**). As previously observed, monkeys generally showed significant choice hysteresis. However, this bias was not increased by electrical stimulation in either time window. The square symbol on the left panel represents one outlier ($y = 3.28$; distance from mean = 5.11 SD), which was removed from statistical analyses. **C**, **D** Choice hysteresis, side and order. Both biases were negligible and neither one was significantly altered by electrical stimulation in either time window. All *p* values were obtained from two-tailed Wilcoxon tests and two-tailed t tests. Source data are provided as a Source Data file.

effect size at weak current exceeded that measured at high current (Fig. 5). The U-shaped trend suggests that increases in choice variability at very low current and at high current were mediated by different cellular mechanisms. Low current stimulation is believed to primarily activate inhibitory interneurons, which are smaller, have higher firing rate, and thus have lower stimulation threshold compared to pyramidal cells[39]. Notably, neuronal recordings in monkeys performing the same choice task used here indicated that decisions rely on circuit inhibition[26,40]. Furthermore, current computational models suggest that value comparison relies on a balance between recurrent excitation and pooled inhibition[6,41,42]. In these models, increasing inhibition makes decisions less accurate[43,44]. Thus we speculate that weak electrical stimulation (≤15 µA) disproportionately increased inhibition and thus disrupted the excitation-inhibition balance, which specifically affected value comparison. Conversely, stimulation at intermediate currents (25–50 µA), which presumably activated both pyramidal cells and interneurons, might have affected choices in other ways while preserving the excitation-inhibition balance. Finally, stimulation at high current (125 µA) might have increased choice variability for several reasons, including by inducing antidromic spikes and/or by affecting fibers of passage. Future work should examine these hypotheses in more detail. Importantly, regardless of the specific cellular mechanisms through which weak electrical stimulation affects neuronal activity, our results indicate that neurons in the primate OFC participate in value comparison.

Leveraging the power of causal manipulation, our results demonstrate that value comparison—i.e., the decision process—engages neurons in OFC and thus a good-based representation of options and values[45]. Low current levels and the absence of spatial signals in the primate OFC strongly suggest that electrical stimulation in our experiments did not affect action planning per se. Confirming this point, reaction times were not altered by stimulation. At the same time, our results do not address the possible role of other brain regions. In other words, we cannot exclude that neurons in some other area, including motor areas, also contribute to value comparison. With this premise, our results are at odds with action-based models, according to which decision making generally involves a competition between possible action plans taking place in motor or premotor regions[9,10]. Conversely, our results are consistent with the good-based model[45] and with the distributed consensus model[11], both of which posit that a broad class of choices takes place partly or fully in good-based representations. (For the present purposes, the good-based model and the distributed consensus model differ in relatively subtle ways. According to the former, economic choice is a distinct mental process categorically distinct from other cognitive functions that may be construed as requiring a choice[45,46]. According to the latter, there is a substantial continuity between different types of choice, all of them engage multiple neuronal representations, and the relative weight of

At higher currents (≥100 µA), electrical stimulation affects cells in more complex ways such as inducing antidromic spikes that can collide with natural spikes (neuronal hijacking)[34–36]. Consequently, the range-dependent bias disappeared[1].

Second, changes in the order bias grew with the current level. All the measures are consistent with the idea that this effect was driven by decelerating response functions[37,38] and/or by neuronal hijacking at high currents[1].

Third, the increase in choice variability had a U-shaped trend: it was observed at very low current (5–15 µA) and at the highest current (125 µA), but not between the extremes (25 and 50 µA). Notably, the

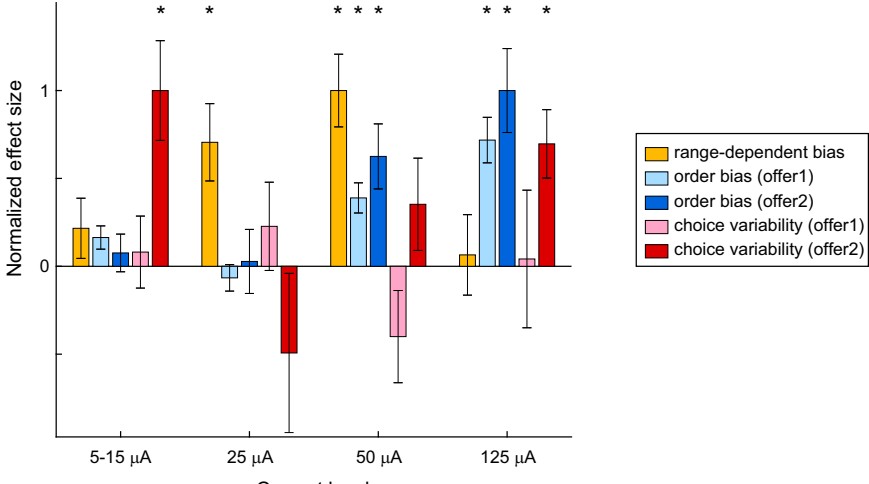

**Fig. 5 | Normalized effect sizes measured at different current levels.** Each effect was rectified and normalized across current levels such that the expected effect was >0 (see "Methods"). The range-dependent bias effect was defined as Pearson's correlation between the change in relative value ($\rho_{\text{stimON}} - \rho_{\text{stimOFF}}$) and the difference in value range ($\Delta V_A - \Delta V_B$). The change in order bias was defined such that the effect would be >0 if electrical stimulation in one time window biased choices in favor of the offer presented in the other time window. For choice variability, the effect was defined >0 if electrical stimulation increased choice variability. Effects were normalized such that the maximum across conditions =1 for each effect. Here histogram bars are population averages, error bars are standard errors, and asterisks indicate statistical significance ($p < 0.01$; two-tailed Pearson's correlation for range-dependent bias; two-tailed Wilcoxon test for all other effects). Exact $p$ values and sample sizes are provided in Supplementary Table 1. For the range-dependent bias, sessions in which stimulation was delivered during offer1 or offer2 were pooled. Data at 5–15 µA are the same as in Fig. 2. Data at 25, 50, and 125 µA are from ref. 1. Source data are provided as a Source Data file.

each representation depends on the type of choice. In particular, choices such as those examined here rely primarily or exclusively on good-based representations[11]).

Our findings are also at odds with models where binary decisions take place in a sequential manner. In particular, the attentional drift diffusion-model (aDDM) holds that decisions are guided by shifts of gaze or visual attention. In this view, while choosing, subjects shift the attentional focus back and forth between the two options; at any given shift, a comparator increments a decision variable in favor of the currently attended offer[13,47]. In our choice task, the subsequent appearance of the two offers at the center of the monitor effectively mimics a gaze shift. If the decision mechanism was that proposed by the aDDM, decisions would be affected by electrical stimulation during either offer1 or offer2. Conversely, the fact that the stimulation induced higher choice variability only when the electric current was delivered during offer2 argues against the aDDM. Along similar lines, our results are hard to reconcile with the proposal that binary choices are constructed through a sequence of accept/reject decisions[15,48]. If this was the case, choices in our experiments should have been affected by electrical stimulation delivered during either offer1 or offer2, contrary to the observations. (Other arguments casting doubts on sequential decisions models were discussed elsewhere[26,49]).

To conclude, in the past 20 years, a series of neurophysiology studies identified in the primate OFC different groups of neurons encoding individual offer values, the binary choice outcome, and the chosen value. The fact that these variables capture both the input and the output of the choice process leads to the hypothesis that these groups of neurons constitute the building blocks of a decision circuit. Experimental and computational results have generally supported this notion, but until recently the evidence remained correlative. Our experiments using electrical stimulation demonstrated that neurons in OFC are causally involved in both the computation and comparison of offer values. These results substantially restrict the domain of defensible hypotheses regarding the neuronal underpinnings of economic choices. In this respect, these results may be viewed as a breakthrough. Looking forward, a major goal will be to shed light on the organization and mechanisms of the decision circuit. Open questions concern the

connectivity between different cell groups in OFC, the connectivity between these cell groups and other brain regions, and the functional role of neuronal inhibition. Future research shall address these fundamental questions.

## Methods

All the experimental procedures conformed to the NIH *Guide for the Care and Use of Laboratory Animals* and were approved by the Institutional Animal Care and Use Committee (IACUC) at Washington University.

### Experimental procedures

The study was conducted on two male rhesus monkeys (*Macaca mulatta*): G (age 8, 9.1 kg) and J (age 7, 10.0 kg). Experimental procedures and data analyses closely resembled those previously described[1]. Before training, we implanted in each monkey a head-restraining device and an oval recording chamber under general anesthesia. During the experiments, the animals sat in an electrically insulated enclosure with their head restrained. A computer monitor was placed 57 cm in front the animal. The behavioral task was controlled through custom-written software (http://www.monkeylogic.net/). The gaze direction was monitored by an infrared video camera (Eyelink; SR Research) at 1 kHz.

The chamber provided bilateral access to OFC. Structural MRIs (1 mm sections) performed before and after surgery were used to guide electrode penetrations. Electrical stimulation focused on the central orbital gyrus, in a region corresponding to area 13/11. During stimulation sessions, low-impedance (100–500 kΩ) tungsten electrodes (100 µm shank diameter; FHC) were advanced using a custom-built motorized micro-drive driven remotely. The stimulating electrode was always positioned well within the gray matter. A second electrode advanced in parallel using the same micro-drive was used to confirm the depth and to record neuronal activity. Stimulation trains were generated by a programmable analog output (Power 1401, Cambridge Electronic Design) and triggered through a TTL by the computer running the behavioral task. Monopolar electric currents were generated by an analog stimulus isolator (Model 2200, A-M Systems).

Electric current was delivered during offer1 or during offer2, in separate sessions. For the present study, stimulation parameters were as follows. Stimulation started 0–100 ms after offer onset and lasted 300–600 ms. The stimulation train was constituted of biphasic pulses (200 μs each pulse, 100 μs separation between pulses) delivered at 125-200 Hz frequency. In different sessions, current amplitudes varied between 5 and 15 μA. Stimulation was performed in both hemispheres of monkey G (left: AP 31:36, ML −7:−12; right: AP 31:36, ML 4:9) and in both hemispheres of monkey J (left: AP 31:35, ML −8:−10; right AP 31:35, ML 6:10) (Supplementary Fig. 2). Electric current was delivered unilaterally or bilaterally, in separate sessions. All the parameters were set at the beginning of each session and were not adjusted within sessions. Similar parameters were used for stimulation at higher currents (≥25 μA; 144 sessions total) [1].

Our present data set includes a total of 91 sessions (58 from monkey G, 33 from monkey J). The number of sessions was not precisely pre-determined at the beginning of the study. Our previous study included 17–29 sessions per condition[1]. Since we expected that the effects of weak stimulation would be subtle, we planned the number of sessions to be in that range for each animal. For offer1 stimulation, we ran a few more sessions to ensure that the lack of significant effects was not due to insufficient statistical power.

### Data analysis: Comparing effect sizes across current levels

Data were analyzed in Matlab (MathWorks Inc). For each session, we examined separately trials with and without the stimulation (stimON, stimOFF). For each group of trials, we analyzed choices using probit regressions. From the fitted parameters, we derived measures for the relative value, the sigmoid steepness, and several choice biases (Eqs. 1–5). A comparison of the behavioral effects of electrical stimulation at different current levels focused on three measures, namely the range-dependent bias, the change in order bias, and the increase in choice variability. Each effect was computed in each session, rectified such that the expected effect was >0, averaged across the relevant population, and normalized across conditions. The normalized effect sizes were thus defined as follows.

(1) *Increase in choice variability*. For each session, we measured the change in sigmoid steepness $\Delta\eta = \eta_{stimON} - \eta_{stimOFF}$. The change in choice variability was defined as $\Delta cv = -\Delta\eta$. Thus $\Delta cv > 0$ indicated that the stimulation increased choice variability. For each time window (offer1, offer2) and for each current level (≤15, 25, 50, and 125 μA), we averaged $\Delta cv$ across the relevant sessions. We thus obtained a measure $\Delta cv_{condition}$ for each of the 8 conditions. Finally, we divided these measures by the maximum, and obtained the normalized effect size shown in Fig. 5.

(2) *Change in order bias*. For each session, we measured the change in order bias induced by the stimulation $\Delta\varepsilon = (\varepsilon_{stimON} - \varepsilon_{stimOFF})$. Suitable stimulation during offer1 (offer2) is expected to increase (decrease) the order bias. Thus we rectified the measures obtained for sessions where stimulation was delivered upon offer2 by changing the sign. For each time window (offer1, offer2) and for each current level (≤15, 25, 50, and 125 μA), we averaged $\Delta\varepsilon$ across the relevant sessions. We thus obtained a measure for $\Delta\varepsilon_{condition}$ for each of the 8 conditions. Finally, we normalized these measures dividing by the maximum, and obtained the normalized effect size shown in Fig. 5.

(3) *Range-dependent bias*. Suitable electrical stimulation increases the firing rates of offer value cells and the effect is equivalent to increasing both offer values. Because of range adaptation, each offer value increases by a quantity proportional to the corresponding value range. As a result, the stimulation is expected to bias choices in favor of the juice offered with the larger value range. More precisely, indicating with $\Delta V_A$ and $\Delta V_B$ the value ranges for juices A and B, respectively, the expected change in relative value $\Delta\rho = \rho_{stimON} - \rho_{stimOFF}$ is proportional to the

difference in value range $\Delta V = \Delta V_A - \Delta V_B$[1]. We previously found that this effect was most pronounced at 50 μA and did not depend on the stimulation window[1]. To generate Fig. 5, we pooled sessions based on the current level (≤15, 25, 50, and 125 μA). For each current level, the effect size was Pearson's correlation r($\Delta\rho$, $\Delta V$) computed across sessions. The four measures were normalized dividing by the maximum effect size across current levels.

All the measures shown in Fig. 5 for current levels ≥25 μA summarize results described in a previous study[1]. Of note, here we use the label "125 μA" instead of "≥100 μA" for clarity, because in the majority of these sessions (47/54) the current was set at 125 μA. In the remaining sessions (7/54), the current was set between 100 and 200 μA.

In each population analysis, we identified as outliers data points that differed from the mean by >3 STD on either axis, and we removed them from the data set. This criterion excluded 1/91 sessions at ≤15 μA and 6/144 sessions at ≥25 μA, as previously described[1], only from the analyses of range-dependent biases. Including these sessions in the analyses did not substantially alter any of the results. Exact *p* values for these analyses are provided in Supplementary Table. 2.

Every trial that did not result in juice delivery was considered an error and included in Fig. 3A. Error trials included failures to initiate the trial (i.e., to acquire the initial fixation), breaks of center fixation at any time prior to the go signal, failure to indicate a choice after the go signal, and breaks of peripheral fixation after target acquisition. Excluding from the analysis failures to the initiate the trial markedly reduced the error rates but did not significantly alter the results of Fig. 3A.

### Reporting summary

Further information on research design is available in the Nature Research Reporting Summary linked to this article.

## Data availability

The supplementary material includes the equivalent of Fig. 1BC for each session in the data set. Raw data are available at: https://github.com/PadoaSchioppaLab/2022_NatComms_lowCurrent Source data are provided with this paper, and can be found at: https://github.com/PadoaSchioppaLab/2022_NatComms_lowCurrent/tree/main/source_data Source data are provided with this paper.

## Code availability

The Matlab code used for the analysis are available at: https://github.com/PadoaSchioppaLab /2022_NatComms_lowCurrent.

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

## Acknowledgements

We thank Bromberg-Martin, E., Conen, K., Monosov, I., Snyder, L., Werginz, P., and members of our lab for helpful comments on previous versions of the manuscript. This research was supported by the National Institutes of Health (grants number R01-DA032758 and R01-MH104494 to CPS).

## Author contributions

S.B. and W.S. collected and analyzed the data; C.P.S. supervised the project and wrote the manuscript. All the authors edited the manuscript.

## Competing interests

The authors declare no competing interests.
