## [Peer Review File · Nature Communications]

Orbitofrontal cortex contributes to the comparison of values underlying economic choicesREVIEWER COMMENTS

Reviewer #1 (Remarks to the Author):

Ballesta and colleagues present a timely study on the neural underpinnings of value comparisons in value selective orbitofrontal cortex.

The study is incredibly important and the rationale well reasoned. I fully agree with the authors that we know much too little about the mechanisms of value comparison in any brain structure and given the authors prior extensive knowledge of the NHPs OFC, this was a well argued place to start.

The authors logic on differentiation of value comparison, value computation and peripheral effects is sound but in my opinion hinges on the certainty that no motor or spatial information is actually encoded within the OFC. The authors provide some basic control analysis that the reviewer agrees with but recent experiments from Schoenbaum and other labs hint at that not being as clear cut as previously assumed.

The reviewer would like to see a more thorough caveat discussion on how the results would be affected if and only if spatial information is indeed present within the simulated and recorded population.

The reviewer believes that the manuscript could be strengthened by further emphasizing the differential effect of stimulation level on the three behavioral effects studied since it nicely illustrates that these behavioral measures are in fact differentiable properly.

All in all I think this is a great study with important implications. I also congratulate the authors for including effect size analysis right away on an NHP ephys paper.

Reviewer #2 (Remarks to the Author):

In this manuscript, Ballesta et al. describe follow-up results to a recent high-profile study, in which they used electrical microstimulation (25-125 uA) to demonstrate a causal role of OFC in value-based decision making. This earlier study clearly demonstrated a role of OFC in representing the subjective value of an option, but the evidence for a direct role of OFC in comparing the value of the options was less clear, because the findings could be interpreted in multiple ways. Here the same authors describe findings

with even weaker microstimulation currents (5-15uA). Interestingly, stimulation with this current only influences the variability of choice, but not other aspects of choice behavior. As the authors note, this is clear evidence for a manipulation of value comparison, without any sign of influencing the value of the options that are to be compared. There is one remaining theoretical possibility for explaining the results, namely a disruption of the motor preparation process by which the desired option will be indicated. The authors mentioning this option and dismiss it, because in primates OFC has no known role in motor generation and the weak current strength makes it unlikely that anatomically distant motor regions are directly affected. I agree with this assessment, although I think the authors should simply state the reasons that make it unlikely, while acknowledging that they have not directly ruled out this possibility. Altogether, the results and the interpretation (a causal role of OFC in value comparison) is clear. This finding is important, because it clarifies the role of OFC in value-based decision making. There are a number of different hypothesis concerning the role of OFC. Hopefully, the new findings will help the field in better distinguishing which is more likely to be correct.

Major concerns:

My main concern is with respect to the mechanistic hypothesis about why weak currents should have the observed highly-specific behavioral effect. The authors do make a valid attempt to discuss possible reasons for the range of effects found with stimulation at different current levels, but in the absence of any supporting evidence, these hypotheses are by necessity speculative. Unfortunately, this contrasts with the conditions for the 'classic' stimulation studies in perceptual decision making. These studies were much easier to understand because in that system, the information on which the decision is based was represented in a cortical area with a well established columnar, map-like organization. Furthermore, the accumulation and comparison of this information was performed in an anatomically separate system. In this context, it would be important to understand the likely anatomical organization of OFC. The authors should provide data about the location and degree of intermixing of the 3 functional types they have distinguished in OFC (both locally within a cortical layer and across different locations of OFC).

Furthermore, I think the authors should address the question to what degree the specific causal roles discovered in OFC are specific to this brain region. Would the authors expect to find similar or different effects for stimulations in other cortical (VMPFC, LPFC, ACC, FEF, SEF, etc.) or subcortical (amygdala, ventral striatum) regions? What would it mean if OFC is part of a larger, more distributed system for decision making?

Lastly, the authors mention a range of very different theories about the role of OFC and the overall functional architecture of decision-making. It would be important to discuss what the meaning of the current findings is for the likelihood that these different theories are correct.

Minor concerns:

- 1) The authors should provide a reconstruction of the locations in OFC that were tested.
- 2) Why is the number of stimulations during offer 1 and offer 2 different?
- 3) What is the definition of 'error' (see Figure 3). Are these aborted trials, no choice trials, choice too early or too late, failure to hold fixation? I noticed that the frequency of these 'errors' is quite high: 15-60% of trials. More data here would be helpful.

4) The authors should provide many more example sessions (either as an extension of Figure 1 or as Suppl. Material). Specifically, I would like to see cases with stronger differences in order bias or steepness during offer 1 presentation. In general, I think the variability of behavioral effects should be illustrated.

Reviewer #1 (Remarks to the Author):

Ballesta and colleagues present a timely study on the neural underpinnings of value comparisons in value selective orbitofrontal cortex. The study is incredibly important and the rationale well reasoned. I fully agree with the authors that we know much too little about the mechanisms of value comparison in any brain structure and given the authors prior extensive knowledge of the NHPs OFC, this was a well argued place to start.

We thank R1 for the overall assessment and for the constructive comments.

The authors logic on differentiation of value comparison, value computation and peripheral effects is sound but in my opinion hinges on the certainty that no motor or spatial information is actually encoded within the OFC. [A] The authors provide some basic control analysis that the reviewer agrees with but recent experiments from Schoenbaum and other labs hint at that not being as clear cut as previously assumed. [B] The reviewer would like to see a more thorough caveat discussion on how the results would be affected if and only if spatial information is indeed present within the simulated and recorded population.

[Brackets are ours]

Thanks for raising these issues. Comments [A] and [B] are closely related, but we address them separately.

[A] Having recorded neuronal and analyzed responses from the primate OFC extensively, including when monkeys performed the same task used here (Ballesta and Padoa-Schioppa, 2019; Shi et al., 2022), we are firmly convinced that the representation of goods and values in our monkeys is non-spatial. However, there is a clear difference between the representation of decision variables in the OFC of primates vs. rodents. This difference emerged early on, when our first paper showing that the encoding of options and values in the monkey OFC was non spatial (Padoa-Schioppa and Assad, 2006) was followed by two reports showing that in the rodent OFC decision variables were encoded in a spatial way (Feierstein et al., 2006; Roesch et al., 2006). Even in our own hands, the nature of the encoding differs across species – indeed, we found that options and values in the mouse OFC were represented in a spatial frame of reference (Kuwabara et al., 2020). On this basis, we think that it is very safe to assume that weak electrical stimulation in the present study did not directly interfere with motor signals.

[B] For the sake of speculation, what if the primate OFC did have some spatial or premotor representation? Would that affect the interpretation of our results? In short, no, it wouldn't. Indeed, if there was a spatial/premotor representation in OFC, we would consider two possible scenarios. (a) One possibility would be that OFC is directly involved in the planning and control of movements. This scenario is very unlikely because OFC does not have direct anatomical connections with any motor area (while cortical motor areas are all interconnected). Furthermore, electrical stimulation in our experiments did not alter response times (**Fig.3**), indicating that the manipulation did not interfere with motor control per se. (b) Alternatively,

the representation of options and values in OFC would be spatial/premotor, yet removed from the very generation and control of movements. In this scenario, our interpretation would be very similar to that presented in the ms. That is, the fact that weak electrical stimulation selectively increases choice variability and does not have other effects on choice biases or response times would indicate that OFC participates in value comparison. The only difference is that this comparison would take place in a spatial representation. Incidentally, action-based models essentially make this proposal (Cisek, 2007; Glimcher et al., 2005), although no one to our knowledge has seriously proposed that action-based value comparison takes place in OFC. In conclusion, our interpretation does not critically depend on the frame of reference in which options and values are represented.

Having said all of this, our current results only demonstrate that OFC participates in value comparison – they do not address the possible role of other brain regions, including motor or premotor regions. Consequently, our results are consistent with both the good-based model (Padoa-Schioppa, 2011) and the distributed consensus model (Cisek, 2012). Both these models posit that a broad class of choices takes place partly or fully in good-based representations, but the latter posits a parallel involvement of action-based representations. In the revised ms, we included a new Discussion section (*Implications for decision models*), where we clarified this point.

The reviewer believes that the manuscript could be strengthened by further emphasizing the differential effect of stimulation level on the three behavioral effects studied since it nicely illustrates that these behavioral measures are in fact differentiable properly.

We completely agree with R1: Comparing the 3 effects at different current levels is very informative, among other reasons because it demonstrates that the 3 effects are independent of each other. As for the emphasis placed on this point, we devote to this issue one full section of the Discussion, one figure and one table. We considered moving the section from the Discussion to the Results, but much of our focus in that section is on the interpretation of the measured effects based on the literature. Thus we think that the section is best placed in the Discussion. However, in response to R1's suggestion, we moved the figure from the supplement to the main text (**Fig.5**).

All in all I think this is a great study with important implications. I also congratulate the authors for including effect size analysis right away on an NHP ephys paper.

Thank you again for these comments.

Reviewer #2 (Remarks to the Author):

In this manuscript, Ballesta et al. describe follow-up results to a recent high-profile study, in which they used electrical microstimulation (25-125 μ A) to demonstrate a causal role of OFC in value-based decision making. This earlier study clearly demonstrated a role of OFC in representing the subjective value of an option, but the evidence for a direct role of OFC in comparing the value of the options was less clear, because the findings could be interpreted in multiple ways. Here the same authors describe findings with even weaker microstimulation currents (5-15 μ A). Interestingly, stimulation with this current only influences the variability of choice, but not other aspects of choice behavior. As the authors note, this is clear evidence for a manipulation of value comparison, without any sign of influencing the value of the options that are to be compared. [1] There is one remaining theoretical possibility for explaining the results, namely a disruption of the motor preparation process by which the desired option will be indicated. The authors mentioning this option and dismiss it, because in primates OFC has no known role in motor generation and the weak current strength makes it unlikely that anatomically distant motor regions are directly affected. I agree with this assessment, although I think the authors should simply state the reasons that make it unlikely, while acknowledging that they have not directly ruled out this possibility. Altogether, the results and the interpretation (a causal role of OFC in value comparison) is clear. This finding is important, because it clarifies the role of OFC in value-based decision making. There are a number of different hypothesis concerning the role of OFC. Hopefully, the new findings will help the field in better distinguishing which is more likely to be correct.

We thank R2 for the overall assessment and for the thoughtful comments.

[1] With respect to the hypothesis that electrical stimulation in our experiments might have affected motor preparation per se, R2 appears to concur with our interpretation while recommending that we inject some word of caution. In response to this and other comments from both reviewers, we added to the Discussion a new section *Implications for decision models*, which includes this passage (p.8):

Low current levels and the absence of spatial signals in the primate OFC strongly suggest that electrical stimulation in our experiments did not affect action planning per se. At the same time, our results do not address the possible role of other brain regions. In other words, we cannot exclude that neurons in some other area, including motor areas, also contribute to value comparison.

Major concerns:

[2] My main concern is with respect to the mechanistic hypothesis about why weak currents should have the observed highly-specific behavioral effect. The authors do make a valid attempt to discuss possible reasons for the range of effects found with stimulation at different current levels, but in the absence of any supporting evidence, these hypotheses are by necessity speculative. Unfortunately, this contrasts with the conditions for the 'classic'

stimulation studies in perceptual decision making. These studies were much easier to understand because in that system, the information on which the decision is based was represented in a cortical area with a well established columnar, map-like organization. Furthermore, the accumulation and comparison of this information was performed in an anatomically separate system. In this context, it would be important to understand the likely anatomical organization of OFC. The authors should provide data about the location and degree of intermixing of the 3 functional types they have distinguished in OFC (both locally within a cortical layer and across different locations of OFC).

The reconstruction of stimulation sites for each animal is now provided in **Fig.S2**. No significant differences were found regarding the location of stimulation sites between offer1 and offer2. As for the intermixing of the 3 functional cell types, we should clarify that recording electrodes were used to ensure stimulation location within OFC for each experimental sessions but we did not bias stimulation location based on neuronal responses. In addition, we did not systematically record from OFC during electrical stimulation sessions. Thus even if the 3 cell groups previously identified in this area did form anatomical clusters, the location of our electrode during stimulation would be blind to, and thus independent of, such putative clustering. Consequently, this putative clustering could not explain our current results. As for whether any such clustering actually exists in OFC, we can only rely on our analysis of previous data sets, where we *did not* find any clustering. On this point, let us quote verbatim a passage of our previous paper (Conen and Padoa-Schioppa, 2015) (p.1376):

... unlike neurons in sensory regions, offer value cells did not appear to cluster based on the encoded variable. Such a clustering would make it more likely to encounter pairs of offer value cells encoding the same variable (same juice, same sign) when two neurons are recorded from the same electrode compared with when two neurons are recorded ≥ 1 mm apart. In contrast, considering all pairs of offer value cells, 15/40 (38%) encoded the same variable when the two neurons were recorded from the same electrode, while 26/73 (34%) encoded the same variable when the two neurons were recorded from different electrodes (Fig. 6, C and D).

[3] Furthermore, I think the authors should address the question to what degree the specific causal roles discovered in OFC are specific to this brain region. Would the authors expect to find similar or different effects for stimulations in other cortical (VMPFC, LPFC, ACC, FEF, SEF, etc.) or subcortical (amygdala, ventral striatum) regions? What would it mean if OFC is part of a larger, more distributed system for decision making?

Both questions [3] and [4] are very important and we thank R2 for raising them. We addressed both of them in the new section *Implications for decision models*. With respect to question [3], we clarified in the ms the following point (p.8):

... our results do not address the possible role of other brain regions. In other words, we cannot exclude that neurons in some other area, including motor areas, also contribute to value comparison.

For what it is worth, our speculation is that motor regions such as ACC, FEF, SEF or LIP *are not* involved in this decision process. For different reasons, our hunch is that none of the LPFC, vmPFC, or striatum plays a direct role in value comparison. Conversely, we view the amygdala as a serious candidate. Indeed we are planning to conduct electrical stimulation experiments focused on the amygdala.

[4] Lastly, the authors mention a range of very different theories about the role of OFC and the overall functional architecture of decision-making. It would be important to discuss what the meaning of the current findings is for the likelihood that these different theories are correct.

The new Discussion section was largely written to address this very question, to which we devote two paragraphs (p.8-9). In essence, our current findings are at odds with action-based models, which hold that decision making generally involves a competition between possible action plans (Cisek, 2007; Glimcher et al., 2005). Conversely, our results are consistent with the good-based model (Padoa-Schioppa, 2011) and with the distributed consensus model (Cisek, 2012). Both these models posit that a broad class of choices takes place fully or partly in good-based representations. Different considerations reveal that our findings are hard to reconcile with models where decisions take place through sequences of steps, such as the attentional drift-diffusion model (Krajbich et al., 2010) and a models where binary choices are conceived as sequences of accept/reject decisions (Hayden and Moreno-Bote, 2018; Kacelnik et al., 2011). Indeed, if the decision mechanism were those proposed by such models, decisions should be affected by electrical stimulation during either offer1 or offer2, contrary to our findings.

Minor concerns:

1) The authors should provide a reconstruction of the locations in OFC that were tested.

The reconstruction of stimulation sites for each animal is now provided in **Fig.S2**.

2) Why is the number of stimulations during offer 1 and offer 2 different?

We did not precisely pre-determine the number of sessions at the beginning of the study. We took as benchmark our previous study (Ballesta et al., 2020), where we had 17-29 sessions per condition (6 conditions). Since we expected that the effects of weak stimulation might be more subtle, we planned for a number of sessions in that range for each animal. For offer1 stimulation, we erred on the side of having more sessions, because we are reporting a negative result and we wanted to be sure that the failure to disprove the null hypothesis was not due to lack of statistical power. We included a paragraph in the Methods to clarify these points.

3) What is the definition of 'error' (see Figure 3). Are these aborted trials, no choice trials, choice too early or too late, failure to hold fixation? I noticed that the frequency of these 'errors' is quite high: 15-60% of trials. More data here would be helpful.

Thanks for asking this question. We added to the Methods the following clarifying paragraph:

Every trial that did not result in juice delivery was considered an error and included in **Fig.3A**. Error trials included failures to initiate the trial (i.e., to acquire the initial fixation), breaks of center fixation at any time prior to the go signal, failure to indicate a choice after the go signal, and breaks of peripheral fixation after target acquisition. Excluding from the analysis failures to initiate the trial markedly reduced the error rates but did not significantly alter the results of **Fig.3A**.

4) The authors should provide many more example sessions (either as an extension of Figure 1 or as Suppl. Material). Specifically, I would like to see cases with stronger differences in order bias or steepness during offer 1 presentation. In general, I think the variability of behavioral effects should be illustrated.

We added as Supplementary Material a PDF including the equivalent of **Fig.1BC** for each session in the data set. We also published online (GitHub) the entire data set.

References

Ballesta, S., and Padoa-Schioppa, C. (2019). Economic decisions through circuit inhibition. *Curr Biol* 29, 3814-3824 e3815.

Ballesta, S., Shi, W., Conen, K.E., and Padoa-Schioppa, C. (2020). Values encoded in orbitofrontal cortex are causally related to economic choices. *Nature* 588, 450-453.

Cisek, P. (2007). Cortical mechanisms of action selection: the affordance competition hypothesis. *Philos Trans R Soc Lond B Biol Sci* 362, 1585-1599.

Cisek, P. (2012). Making decisions through a distributed consensus. *Curr Opin Neurobiol* 22, 927-936.

Conen, K.E., and Padoa-Schioppa, C. (2015). Neuronal variability in orbitofrontal cortex during economic decisions. *J Neurophysiol* 114, 1367-1381.

Feierstein, C.E., Quirk, M.C., Uchida, N., Sosulski, D.L., and Mainen, Z.F. (2006). Representation of spatial goals in rat orbitofrontal cortex. *Neuron* 51, 495-507.

Glimcher, P.W., Dorris, M.C., and Bayer, H.M. (2005). Physiological utility theory and the neuroeconomics of choice. *Games Econ Behav* 52, 213-256.

Hayden, B.Y., and Moreno-Bote, R. (2018). A neuronal theory of sequential economic choice. *Brain and Neuroscience Advances*, 1-15.

Kacelnik, A., Vasconcelos, M., and Monteiro, T. (2011). Darwin's "tug-of-war" vs. starling's "horse-racing": how adaptations for sequential encounters drive simultaneous choice. *Behav Ecol Sociobiol* 65, 547-558.

Krajbich, I., Armel, C., and Rangel, A. (2010). Visual fixations and the computation and comparison of value in simple choice. *Nat Neurosci* 13, 1292-1298.

Kuwabara, M., Kang, N., Holy, T.E., and Padoa-Schioppa, C. (2020). Neural mechanisms of economic choices in mice. *Elife* 9.

Padoa-Schioppa, C. (2011). Neurobiology of economic choice: a good-based model. *Annu Rev Neurosci* 34, 333-359.

Padoa-Schioppa, C., and Assad, J.A. (2006). Neurons in orbitofrontal cortex encode economic value. *Nature* 441, 223-226.

Roesch, M.R., Taylor, A.R., and Schoenbaum, G. (2006). Encoding of time-discounted rewards in orbitofrontal cortex is independent of value representation. *Neuron* 51, 509-520.

Shi, W., Ballesta, S., and Padoa-Schioppa, C. (2022). Economic choices under simultaneous or sequential offers rely on the same neural circuit. *J Neurosci* 42, 33-43.

REVIEWERS' COMMENTS

Reviewer #1 (Remarks to the Author):

The authors have fully addressed all my concerns.

I congratulate the authors for a wonderful manuscript and highly important addition to the literature.

Reviewer #2 (Remarks to the Author):

The authors have responded very satisfactory to my minimal concerns in my initial review. I think the changes have improved the manuscript, and I think it is ready to be accepted.

REVIEWERS' COMMENTS

Reviewer #1 (Remarks to the Author):

The authors have fully addressed all my concerns.

I congratulate the authors for a wonderful manuscript and highly important addition to the literature.

Thank you!

Reviewer #2 (Remarks to the Author):

The authors have responded very satisfactory to my minimal concerns in my initial review. I think the changes have improved the manuscript, and I think it is ready to be accepted.

Thank you!